# Effect of High-Intensity Ultrasound Pretreatment on the Properties of the Transglutaminase (TGase)-Induced β-Conglycinin (7S) Gel

**DOI:** 10.3390/foods12102037

**Published:** 2023-05-17

**Authors:** Lan Zhang, Jixin Zhang, Pingping Wen, Jingguo Xu, Huiqing Xu, Guiyou Cui, Jun Wang

**Affiliations:** Tourism and Cuisine College, Yangzhou University, Yangzhou 225127, China; mx120211221@stu.yzu.edu.cn (L.Z.); mx120221256@stu.yzu.edu.cn (J.Z.); mx120201185@stu.yzu.edu.cn (P.W.); 008122@yzu.edu.cn (J.X.); gycui@yzu.edu.cn (G.C.); 007232@yzu.edu.cn (J.W.)

**Keywords:** ultrasound pretreatment, ultrasound time, soy proteins, β-conglycinin, protein structure, transglutaminase-induced gel, gelling properties

## Abstract

In this study, we investigated the effects of different high-intensity ultrasound (HIU) pretreatment times (0–60 min) on the structure of β-conglycinin (7S) and the structural and functional properties of 7S gels induced by transglutaminase (TGase). Analysis of 7S conformation revealed that 30 min HIU pretreatment significantly induced the unfolding of the 7S structure, with the smallest particle size (97.59 nm), the highest surface hydrophobicity (51.42), and the lowering and raising of the content of the α-helix and β-sheet, respectively. Gel solubility showed that HIU facilitated the formation of ε-(γ-glutamyl)lysine isopeptide bonds, which maintain the stability and integrity of the gel network. The SEM revealed that the three-dimensional network structure of the gel at 30 min exhibited filamentous and homogeneous properties. Among them, the gel strength and water-holding capacity were approximately 1.54 and 1.23 times higher than those of the untreated 7S gels, respectively. The 7S gel obtained the highest thermal denaturation temperature (89.39 °C), G′, and G″, and the lowest tan δ. Correlation analysis demonstrated that the gel functional properties were negatively correlated with particle size and the α-helix, while positively with Ho and β-sheet. By contrast, gels without sonication or with excessive pretreatment showed a large pore size and inhomogeneous gel network, and poor properties. These results will provide a theoretical basis for the optimization of HIU pretreatment conditions during TGase-induced 7S gel formation, to improve gelling properties.

## 1. Introduction

Soy protein is used as a crucial food processing aid because it contains high-quality protein required by the human body and is a great source of essential fatty acid and isoflavone [1]. In addition, it has a superior ability to improve the texture of food. According to statistics, the global production of soy protein was estimated to be 392 million tons in 2022. Gel products are the largest component, with a market share of 41% [2]. The β-conglycinin (7S), an important component of soy protein, is a trimer composed of three subunits (α′, α, β), linked by non-covalent interactions [3]. The study has shown that 7S has excellent physiological functions, such as enhancing LDL metabolism and lowering triglycerides in human hepatocytes [4]. In addition, its application function—gelling properties—has potential in a variety of fields, e.g., food industries such as meat and dairy products [5,6,7], and biomedical industries such as drug delivery and biosensors [8]. Nevertheless, the native 7S exhibited poor gel properties. Feng, et al. [9] stated that the conventional thermally induced 7S gels had a loose and brittle structure and exhibited poor functional properties. Kohyama et al. [10] also mentioned that 7S gels possessed lower levels of fracture stress values and higher shrinkage, which is not favorable for the practical application of the gels. Therefore, in order to solve the related problems, many modification methods such as chemical (e.g., glycosylation, acylation, and deamidation), physical (e.g., high static pressure, ultrasound treatment, and extrusion cooking), and biological (e.g., enzymes and fermenting bactreia) have been attempted [11]. In the above modification strategy, the enzyme-promoted reactions result in profound changes in the protein gel properties and structural properties that occur without altering the flavor, color, and nutrition of the food [12]. Therefore, it is widely applied in protein modification, with transglutaminase being the most utilized.

Transglutaminase (TGase) catalyzes the acyl transfer reaction between glutamine residues (Gln) and lysine residues (Lys) in the protein chain to form ε-(γ-glutamyl) lysine isopeptide bonds, which are eventually cross-linked inter- or intramolecularly by covalent cross-linking [7,13]. Sun, et al. [14] demonstrated that the strength of pea protein and soy protein gels induced with TGase was 8-fold and 2-fold higher than that of the control, respectively. Similarly, Chen, et al. [15] and Qin, et al. [16] noted that the addition of TGase significantly promoted the formation of fine gel structures of rapeseed isolate and wheat glutenin, which in turn improved the gel properties. It has been shown that the active site of TGase is readily accessible for most subunits of 7S, which facilitates protein–enzyme cross-linking, allowing 7S to serve as an excellent TGase cross-linking substrate and facilitating further improvement of its gel properties [17]. However, both Gln and Lys are encapsulated inside the native 7S molecule, which does not facilitate cross-linking with TGase. Thus, proper pretreatment of 7S is required to expose the internal groups and to enhance protein–protein and protein–enzyme reactions [5,18].

Ultrasonic applications in the food industry are gaining more and more attention from researchers. Modifications of food products by sonication technology include low-intensity ultrasound pretreatment and high-intensity ultrasound pretreatment. The low-intensity ultrasound (5–10 kHz, 1 W/cm^−2^) is mostly used for the diagnosis and analysis of foodstuff [19]. The high-intensity ultrasound (20–100 kHz, 10–1000 W/cm^−2^) is used to change the physical and chemical properties of food or material, to affect their functionality [20]. High-intensity ultrasound (HIU) is used extensively in food processing because of its ease of operation, relative economy, and excellent modification effects on substances, such as degassing, emulsification properties improvement, ingredient extraction, and food sterilization [21]. Meanwhile, the effect of HIU treatment on protein property has attracted the attention of many researchers. For example, whey [22] and quinoa [23] protein showed significant unfolding of protein structures and the increasing solubility of the protein dispersions after sonication. Li, et al. [24] found that the α-helix and β-sheet content in the protein secondary structure was decreased and increased, respectively, which was related to the de-folding of the protein conformation, when they prepared peanut isolate-polysaccharide couples by ultrasonication combined with glycosylation reaction. The textural characteristics of soy protein gels induced by CaSO_4_ [6], glucono-δ-lactone [25], and acid [26] were all improved after HIU treatment, which could relate to the significant perfection of gelation structure. In addition, Tang, et al. [27] noted that HIU pretreatment can convert insoluble precipitates into soluble aggregates, which in turn significantly improves the gel properties. Zhang, et al. [28] reported that sonication caused changes in the peptide backbone conformation of proteins, allowing exposure of buried internal lysine residues and increasing accessibility of TGase to protein molecules. Based on these studies, we hypothesized that HIU pretreatment has promise in improving the functional properties of TGase-induced 7S gels. Currently, most studies are focused on the modification of whey protein, surimi protein, and beef protein, and the modification of 7S is still relatively rare. Furthermore, there is a lack of information on the performance of HIU pretreatment combined with TGase-modified 7S gels.

Therefore, this study investigated the effect of different HIU pretreatment times on the 7S structure and the gel properties induced by TGase. The changes in 7S conformations (primary, secondary, tertiary, and quaternary structures) with different HIU pretreatment times were compared. The differences in structural properties as well as functional properties of gels possessing these structural variations of protein were explored. The correlation between the protein conformation and the functional properties of the gel was analyzed. The results of this study will provide theoretical and technical support for the development of high-quality 7S gel products, and facilitate the development of new applications of 7S in the food processing industry.

## 2. Materials and Methods

### 2.1. Materials and Regents

Defatted soy flour was purchased from Shandong Yuwang Industrial Co., Ltd. (Yucheng, China). TGase was obtained from the Bolian Biotechnology Food Co., Ltd. (Zhengzhou, China). Tris-base, Glycine, sodium dodecyl sulfate (SDS) was obtained from Bioengineering (Shanghai) Co., Ltd. (Shanghai, China). Kormas Brilliant Blue R-250, N, N, N, N-tetramethyl-ethylenediamine (TEMED) and β-mercaptoethanol were purchased from Maclean Biochemical Co., Ltd. (Shanghai, China).

### 2.2. Preparation of 7S Protein

The 7S was prepared based on the description of Liu, et al. [29], with some modifications. Briefly, defatted soybean flour was extracted with 0.03 M Tris-HCl buffer (pH 8.5) and dispersed at a flour: Tris-HCl ratio of 1:15 (*w/v*) at 45 °C for 1 h, after which the dispersion was centrifuged (9000× *g*, 30 min, 4 °C) using the Allegra X-30R Centrifuge (F0850 rotor, Beckman Coulter, Inc., Bria, CA, USA). The supernatant was added sodium bisulfite to a concentration of 0.98 g/L and the pH was adjusted to 6.4 using 2 M HCl (subsequent pH adjustments were carried out using 2 M HCl, unless otherwise specified). The above solution was stored overnight at 4 °C and subsequently centrifuged (6500× *g*, 20 min, 4 °C). After that, the supernatant was collected. NaCl was added to the supernatant (0.25 mol/L), the pH was adjusted to 5.5, and the mixture was stirred for 30 min at room temperature (RT). After stirring, the mixture was centrifuged (9000× *g*, 30 min, 4 °C) and the supernatant was collected. The same volume of deionized water was added to the supernatant. The pH of the resulting solution was adjusted to 5.0 and the solution was centrifuged (6500× *g*, 20 min, 4 °C). The precipitate was washed 3 times with distilled water (DIW) and the pH was adjusted to 7.5 with 2M NaOH. After that, the precipitate was freeze-dried using a Telstar (Lyoquest-55) laboratory freeze dryer (Nanjing Xinfeida Optoelectronics Science and Technology Co., Ltd., Nanjing, China) for 48 h at −55 °C to obtain the 7S.

### 2.3. High-Intensity Ultrasound Pretreatment

The 7S solution (10.0%, *w/v*) was prepared by redissolving the 7S powder obtained from Section 2.2 into DIW, followed by gentle agitation for 2 h at RT, and stored at 4 °C overnight. The solution was ultrasonicated in the flat-bottom conical flask, using a 40 kHz, 200 W GA-3200SP ultrasound machine (Wuxi Shangjia Biological Technology Co., Ltd., Wuxi, China) for different times (0, 15, 30, 45, and 60 min). Throughout the HIU pretreatment phase, the flat-bottom conical flask containing the solution was placed in an ice-water bath. Then, the HIU-treated samples were freeze-dried and stored in a desiccator at RT.

### 2.4. Determination of Particle Size

The 7S powder obtained from Section 2.3 was redissolved in DIW containing 0.01% sodium azide to the 7S concentration of 10% and stirred for 2 h at RT. The solution was preserved for 12 h at 4 °C to ensure complete dissolution of 7S. The particle size of the 7S solution was measured using a Zetasizer Nano ZS analyzer (Shanghai Baiji Instrument System Co., Ltd., Shanghai, China).

### 2.5. Determination of Surface Hydrophobicity (Ho)

The 1-anilino-8-naphthalene-sulfonate (ANS) was applied to measure Ho [30]. The 7S samples were diluted with phosphate buffer (0.01 M, pH 7.0) into a range of protein concentrations (from 1 to 0.15 mg/mL). A total of 60 μL of ANS was added to 4 mL protein solutions at different concentrations. The mixture was incubated for 15 min, protected from light. An F320 fluorescence-spectrophotometer (Shuangxu Electronics Co., Ltd., Shanghai, China) with 365 nm (excitation) and 484 nm (emission) was used to measure fluorescence intensity. The Ho was obtained by calculating the initial slope of fluorescence intensity versus protein concentration (mg/mL) (obtained by linear regression analysis).

### 2.6. Sodium Dodecyl Sulfate-Polyacrylamide Gel Electrophoresis (SDS-PAGE) Analysis

SDS-PAGE was assessed using a 4% stacking gel and a 12% separation gel following the method described by Sun, et al. [31]. The VE-180 Mini Vertical Electrophoresis Cell Device (Tanon Science & Technology Co., Ltd., Shanghai, China) was applied to perform vertical electrophoresis. The gel was stained using Coomassie Brilliant Blue G-250 and then decolorized overnight using methanol/acetic acid/water (1:1:8, *v/v/v*).

### 2.7. Determination of Intrinsic Fluorescence Spectra

The fluorescence spectra of tryptophan was measured based on Li, et al. [32]. The 7S sample was thinned to 1 mg/mL with a phosphate buffer (0.01 M, pH 7.0). The 3 mL of the dispersion was placed in a quartz cell using an F320 fluorescence-spectrophotometer (Shuangxu Electronics Co., Ltd., Shanghai, China). The spectrum obtained with a scanning speed was 40 nm/s at the excitation and emission wavelength of 280 nm and 300–500 nm, respectively.

### 2.8. Circular Dichroism (CD) Analysis

At a temperature of 25 °C, the CD spectra of the 7S samples were obtained using a J-810 CD spectroscopic polarimeter (JASCO Corporation, Tokyo, Japan) for both far- and near-UV CD. The 7S samples were dissolved in DIW and adjusted to concentrations of 0.25 mg/mL and 1 mg/mL for the far- and near-UV measurements, respectively. The far-UV CD spectra were recorded from 190 nm to 260 nm with a scan rate of 100 nm/min, while near-UV CD spectra were recorded from 250 nm to 320 nm with the same scan rate. Both measurements were carried out in quartz cuvettes with a 2 mm path length. Every data point presented here is the average of three independent experiments after correcting for the spectrum of DIW. The collected data were analyzed by an online Circular Dichroism Website: http://dichroweb.cryst.bbk.ac.uk (accessed on 20 October 2022).

### 2.9. Transglutaminase-Reduced β-Conglycinin Gel Formation

The samples prepared as described in Section 2.3 were dispersed in deionized water and stirred at 25 °C for 2 h to obtain 10% (*w/v*) 7S solutions. TGase (enzyme/protein ratio of 6 mg enzyme/100 mg protein) (Bolian Biotechnology Co., Ltd., Lishui, China) was added and stirred evenly. After that, the protein solutions were set at 55 °C for 1 h. Following this, the solutions were placed in water bath at 95 °C for 15 min to inactive the TGase, and were immediately cooled by tap water to form gels. The gels were stored at 4 °C for 24 h before carrying out the subsequent investigations.

### 2.10. Determination of Free Sulfhydryl (FSH) and Total Sulfhydryl (TSH) Contents of Gel

The FSH and TSH contents of the gels were measured using the methodology outlined by Beveridge et al. [33], with minor modifications. The level of the sulfhydryl (SH) groups was determined using 5,5′-dithiobis (2-nitrobenzoic acid) (DTNB). To measure the FSH content, the gels were dissolved in a buffer B (0.086 M Tris, 0.09 M glycine, and 4 mM Na_2_EDTA, pH 8.0), while BSU (B containing 0.5% sodium dodecyl sulfate and 6 M urea) was used to measure the TSH content. The gels were homogenized with buffer at a ratio of 0.1 g protein per 50 mL buffer, before being centrifuged at 10,000× *g* for 20 min at 4 °C. The resulting supernatant fractions were analyzed for FSH and TSH content. The 30 μL Ellman’s reagent solution (10 mM DTNB) was added to 3 mL of the supernatant, which was then mixed quickly and allowed to stand for 15 min at RT. The absorbance was measured at 412 nm (A_1_). The buffers were used instead of protein solutions and Ellman’s reagent as a reagent blank (A_2_) and a protein blank (A_3_), respectively. The content of the SH group was evaluated by the following equation:(1)$$\mathrm{SH}\;\mathrm{group}\;\left(\mu \mathrm{mol}/g\;\mathrm{protein}\right)=\frac{(A_{1}-A_{2}-A_{3})}{13,600\times b\times c}\times 10^{6}$$
where 13,600 is the molar extinction coefficient of Ellman’s reagent, b is the length of the light path (0.5 cm), and c is the protein concentration (mg/mL).

### 2.11. Determination of the Gel Solubility

The solubility of the gels was measured following the method reported by Gómez-Guillén et al., with slight modifications [34]. Specifically, 0.6 M NaCl (S1), 0.6 M NaCl + 8 M urea (S2), and 0.6 M NaCl + 8 M urea + 0.5 M β-mercaptoethanol (S3) were chosen, according to their different ability to break chemical bonds. The solubility of gels in these solutions allowed us to determine the presence of non-specific associations (protein dissolved in S1), hydrogen bonds and hydrophobic interactions (differences in protein content dissolved in S2 and S1), and disulfide bonds (differences in protein content dissolved in S3 and S2). To carry out this measurement, 0.5 g of gel was added to 5 mL of each solution, homogenized at 10,000 rpm for 1 min, and then centrifuged at 10,000× *g* rpm for 30 min at 4 °C. The supernatant was used to determine protein levels (mg soluble protein/mL of homogenate) using the P0006C Bradford Protein Concentration Determination Kit (Biyuntian Biotechnology Co., Ltd., Shanghai, China).

### 2.12. Scanning Electron Microscopy (SEM) Analysis

Small gel pieces (2 × 5 mm) were fixed with 2.5% glutaraldehyde (pH 7.2) at 4 °C for 2 h, and subsequently washed three times in phosphate buffer (0.1 M, pH 7.2). The ethanol with different concentrations of 50%, 70%, 80%, 90%, and 100% was used to dehydrate, and each dehydration was maintained for 10 min. The gel pieces were then vacuum freeze-dried. Their microstructure was observed using an S-4800II SEM (Hitachi, Ltd., Tokyo, Japan). Digital images were obtained at an accelerating voltage of 10 kV and 2000× magnification [35].

### 2.13. Determination of Water-Holding Capacity (WHC)

The gel samples were centrifuged (8000× *g*, 20 min, 4 °C). The water was removed after centrifugation. The *WHC* was calculated using the following formula [36]:(2)$$WHC\;\left(\%\right)=\frac{m-\left(m_{1}-m_{2}\right)}{m}\times 100$$
where *m* (g) is the total mass of the gel before centrifugation, *m*_1_ (g) is the total mass of the centrifuge tube and gel before centrifugation, and *m*_2_ (g) is the total mass of the centrifuge tube and gel after centrifugation.

### 2.14. Determination of Gel Strength

Gel strength was assessed using the method of Zhao, et al. [37], whereby cylindrical samples measuring 20 mm in height and diameter were cut. The food property analyzer (TMS-PRO, FTC Company, Sterling, VA, USA) was utilized to measure the gel strength, with the probe being inserted into the gel at a constant rate of 1 mm/s. The trigger force and drop height were 3 g and 10 mm, respectively.

### 2.15. Differential Scanning Calorimetry (DSC) Analysis

Differential scanning calorimetry was recorded by a DSC apparatus (DSC 8500, PerkinElmer, Inc., Waltham, MA, USA) with a heating rate of 10 °C/min and temperature range of 26–120 °C. In each hermetic aluminum pan, a 5 mg gel sample was sealed, and an empty pan was used as reference.

### 2.16. Rheological Measurements

Following the method and parameter described by Wu, et al. [38], with minor modifications, a Kinexus Pro rotational rheometer equipped with parallel plates (40 mm diameter and 1 mm gap) was used to test the dynamic rheology of the gel dispersion sample. The gel dispersion sample was heated from 25 °C to 95 °C and cooled back to 25 °C. The frequency scan was performed by applying 0.01% strain at 25 °C and scanning in the range of 0.1–10 Hz.

### 2.17. Statistical Analysis

Triplicate measurements were taken and expressed as mean ± standard deviation (SD). The data were analyzed using analysis of variance (ANOVA) on SPSS version 19.0 software (SPSS Institute, Chicago, IL, USA) and the significance tested using Duncan’s multiple range test at *p* < 0.05. The correlation heat map of Pearson coefficient (r) and significance (*p*) was established using Origin2021 (OriginLab Institute, Northampton, MA, USA).

## 3. Results and Discussion

### 3.1. Effect of HIU Pretreatment Time on Conformational Characteristics of 7S

#### 3.1.1. Particle Size

The functional properties of protein are influenced by various factors, including particle size [25]. After 30 min of HIU pretreatment, the particle size of 7S was reduced from 114.4 nm to 97.6 nm (*p* < 0.05), as shown in Figure 1A. This reduction could be attributed to the dissociation between 7S molecules caused by the cavitation forces and the shear energy generated by the accompanying HIU pretreatment [6]. As the HIU time increased, the particle size of 7S slightly increased from 97.6 nm to 108.4 nm, due to the recombination of protein molecules through non-covalent interactions (e.g., electrostatic interactions and hydrophobic interactions) [27]. It has been shown that the longer HIU time caused the formation of small aggregates between protein molecules, leading to an increase in particle size [5,39]. Additionally, there was a slight thermal denaturation of the protein, which also contributed to the variation in particle size [40].

#### 3.1.2. Surface Hydrophobicity (Ho)

The Ho is related to the number of hydrophobic groups of the polar water environment to which the protein surface is exposed, and is an important indicator for the assessment of the process of gel formation by 7S protein. The Ho increased significantly at the initial 30 min (*p* < 0.05) and decreased significantly at longer HIU pretreatment (>30 min) (*p* < 0.05) compared with the native 7S (Figure 1B). The increment in Ho was caused by a certain degree of expansion and stretching of 7S molecules caused by sonication, which exposed the hydrophobic amino acids previously buried inside the 7S molecules. These groups interconnect the protein molecules through hydrophobic interactions during later gel formation, facilitating the gel network improvement [5,41]. The results were in accordance with those reported by Cui, et al. [42]. However, the Ho of the 7S was reduced after HIU pretreatment for more than 30 min. This phenomenon is attributed to the imbalance between the two, resulting from excessive sonication: the exposure levels of the hydrophobic group and the reaggregation levels through hydrophobic interaction [41]. Additionally, one study revealed that the Ho of whey protein was decreased at a sonication time longer than 5 min [43]. Deviations from the present experimental results could be due to differences in experimental material and pretreatment conditions.

#### 3.1.3. SDS-PAGE

SDS-PAGE profiles could effectively present the changes of 7S protein subunits at different HIU pretreatment times. Figure 2 shows that there was no significant difference in the protein subunits at different HIU time. This indicated that different sonication times did not cause changes in the primary structure of 7S and that the sonication treatment also did not lead to the dimerization of 7S. Hu, et al. [44] also obtained a similar phenomenon. However, the increase in particle size and Ho was found in Section 3.1.1, which suggested that aggregation between protein molecules might occur through non-covalent interactions (e.g., electrostatic interactions, hydrophobic interactions, and hydrogen bonds), rather than covalent interactions, such as intermolecular disulfide bridges [5].

#### 3.1.4. Intrinsic Fluorescence Spectra

Quantification of protein molecule unfolding was achieved by the fluorescence intensity detection of residual sites within the protein. The tryptophan was usually used as a probe to reflect the polarity of the local environment of the protein [45]. The maximum fluorescence of tryptophan fluorescence intensity decreased from 24.785 to 22.953 with the increasing time of HIU (0–30 min) (Figure 3A). Further pretreatment increased the maximum fluorescence intensity to 24.592. These results implied that the HIU pretreatment (<30 min) promoted 7S structure unfolding, leading to the increased exposure of chromogenic groups to the solvent and the consequent interaction with quenching agents, ultimately resulting in decreased tryptophan fluorescence intensity [46]. Furthermore, the higher fluorescence intensity with the longer HIU pretreatment time (>30 min) was probably due to HIU pretreatment disrupting the 7S molecular structure, and the 7S underwent coiling and folding, which weakened the fluorescence quenching [47]. Additionally, the fluorescence spectroscopy of 7S allowed for the assessment of tertiary structure changes in 7S molecules.

Sonication for 15 and 30 min produced a minor red shift in the fluorescence emission maximum wavelength (λ_max_) compared to native 7S, from 330.4 nm to 330.8 and 331 nm, respectively. It was inferred that HIU pretreatment (<30 min) exposed more tryptophan to a more polar microenvironment by de-folding 7S. After 45 and 60 min, there was a blue shift in λmax from 331 nm to 330.6 and 330.2 nm, intimating the transfer of tryptophan residues into the internal microenvironment of 7S and forcing the tryptophan residues to be buried by protein aggregates. Overall, the HIU pretreatment changed the structural properties of 7S, and the 7S of 30 min HIU pretreatment had a looser tertiary conformation, compared to other groups.

#### 3.1.5. Circular Dichroism (CD) Spectra

The far-UV CD spectra is a sensitive and informative tool to reflect the secondary structure of the protein side chain. Various secondary structures have their unique spectral characteristics, but the location of their absorption peaks varies, depending on various factors (e.g., protein type, buffer system, and protein concentration) [48]. Typically, the distributions of the secondary structure are as follows: α-helix (193, 222, and 208 nm), β-sheet (195 and 218 nm), β-turn (180–190 and 206 nm), and random coil (195 and above 210 nm) [49]. As shown in Figure 3B, the 7S without HIU pretreatment exhibited prominent features in the following three bands: 192 nm (positive band), 197 nm (over-zero point), and 208 nm (negative band). After HIU pretreatment, the excess zero moved toward nearly 200 nm, while the ellipticity value increased at 208 nm. The above results indicated that the decrease in α-helix content and increase in β-sheet content were seen after HIU pretreatment. Notably, the 7S with 30 min HIU pretreatment had smaller negative ellipticity at 208 nm and 222 nm, illustrating less α-helix compared to other groups (Figure 3B). The above phenomenon is visually reflected in Figure 3C, where the α-helix of 7S after 30 min treatment reached its lowest. The present study was in line with the results of Liu, et al. [50], noting that the formation of other secondary structures may be associated with the loss of α-helix.

Figure 3C further illustrates the proportional changes in the secondary structure of 7S. The percentage of β-turn and random coil structure remained relatively stable with the increasing time of HIU. Concurrently, the level of the α-helix decreased gradually within the initial 30 min of HIU pretreatment, while the level of the β-sheet increased accordingly. Nevertheless, further HIU pretreatment caused a slight rise in the α-helix content and a decrease in β-sheet content. These results indicated that optimal sonication could effectively unfold the 7S molecular structure, facilitating the exposure of hydrophobic regions. Excessive sonication, however, caused protein molecules to aggregate into clusters. A similar phenomenon was observed in the study conducted by Liu, et al. [51] when HIU pretreatment was performed on hemp protein isolate. Wu, et al. [52] also revealed that a decrease in α-helix content might correspond to protein unfolding, while an increase in β-sheet content might be a precondition of the formation of polymer and cross-linked gel networks. Proper HIU pretreatment induced more protein-unfolding behavior and exposed protein hydrophobic groups, which was corroborated with changes in Ho (Section 3.1.2). According to the report, the unfolding of the structure enhanced the possibility of protein-to-enzymatic crosslinking, and promoted the protein–enzyme interaction, which further contributed to improving the strength of protein intermolecular forces [16]. Finally, the 7S skeleton became more stable and the 7S gel structure became more ordered.

Near-UV CD spectra can probe the differences of tertiary and/or quaternary conformations in protein, to depict three aromatic amino acid residues. Among the wavelengths of 290–305 nm, tryptophan (Trp) displays a distinct structure, while tyrosine (Tyr) exhibits a fine structure within 275 and 282 nm, and phenylalanine (Phe) shows a fine structure from 255 to 270 nm [48]. Because the Trp band magnitude changes were similar in these 7S samples (Figure 3D), and were consistent with those observed in the endogenous fluorogram (Figure 3A), the changes in Tyr and Phe were the main reason for the alterations in tertiary and/or quaternary structures. The Tyr band magnitude was significantly lower for 7S treated at 30 min, indicating the higher flexibility of the quaternary conformation than other groups. When HIU pretreatment was applied for 15 and 60 min, subtle differences occurred in the near-UV spectra, compared to native 7S. However, the band intensity decreased slightly at 45 min and decreased significantly at 30 min. HIU pretreatment caused different degrees of expansion of the protein tertiary structure, resulting in hydrophobic amino acid residue being exposed to the different levels of the polar microenvironment [53]. This phenomenon explained the varying degree of attenuation observed in the near-UV CD spectra band of 7S with increasing HIU time, which was supported by the findings of Ma, et al. [54].

### 3.2. Structure Properties of TGase-Induced 7S Gels

#### 3.2.1. Free Sulfhydryl (FSH) and Total Sulfhydryl (TSH) Contents of Gel

SH groups can form S–S bonds in the process of gel formation through oxidation reaction [55]. Therefore, we measured the SH group content in the gel to indirectly reflect the change in S–S bond content in the gel, which is closely related to the gel network structure and mechanical properties. Figure 4A shows the SH content of TGase-induced 7S gel with different HIU pretreatment times. The content of FSH and TSH in gels showed a decreasing trend with increasing HIU pretreatment time, but not significant (*p* > 0.05), which was similarly noted by the results of Huang, et al. [56] and Yu Yang [57]. The reduction of the above TSH and FSH may be due to the unfolding of the 7S molecule caused by HIU, exposing the SH residues within the structure to the surface and increasing the possibility of disulfide bond formation during gelation [5,25,39]. However, the lower SH content in 7S eventually led to an insignificant reduction of sulfhydryl groups [58].

The proportion of FSH to TSH is related to the level of protein unfolding or denaturation by external action, where a higher proportion corresponds to a greater level of protein structure unfolding [57]. Compared to the untreated group, the ratio of HIU pretreatment gels was significantly lower (*p* < 0.05), indicating that HIU pretreatment resulted in a weaker degree of unfolding of the tertiary structure of 7S cross-linked by TGase. The decline in the ratio may be attributed to the increased accessibility of TGase to protein caused by HIU pretreatment, leading TGase to inhibit the amplification of the 7S molecule during the induction of gel formation.

#### 3.2.2. Solubility of Gel

The interactions between protein molecules, such as ionic bonds, hydrogen bonds, disulfide bonds, and hydrophobic interactions, play a crucial role in the formation and maintenance of the gel network structure [59]. In order to reveal the impact of different sonication times on the molecular forces of TGase-induced 7S gel, the solubility of the gel in three different denaturing agents was analyzed. The denaturing agents used were S1 (0.6 M NaCl solution), which breaks ionic bonds; S2 (S1 + 8 M urea), which breaks hydrogen bonds and prevents the formation of hydrophobic interaction; and S3 (S2 + 0.5 M β-mercaptoethanol), which breaks disulfide bonds. The solubility of the 7S gel samples in these denaturing solvents is shown in Figure 4B. It was clear from the results that the combination of ionic bonds, hydrogen bonds, hydrophobic interaction, and disulfide bonds formed and stabilized the 7S gel network structure (solubility: S1 < S2 < S3). Moreover, the gel solubility in (S2–S1) was relatively higher, while slight changes in solubility were observed in solvent (S3–S2). The above results indicated that the contribution of hydrogen bonds and hydrophobic interaction was vital in maintaining the gel structure, whereas the contribution of disulfide bonds was limited, which was mutually complementary with the changes in SH group levels observed in Section 3.2.1. The solubility of the 7S gel in the three solvents reached its minimum at 30 min HIU pretreatment time. This result could be attributed to the following main factor: The proper HIU pretreatment facilitated the protein molecules to be unfolded, exposed the buried enzymatic reaction sites inside the molecules, and maximally strengthened the catalysis of TGase to 7S. Eventually, more ε-(γ-glutamyl)lysine isopeptide bonds were formed between glutamate and lysine in 7S molecules, which exert a 20-fold higher strength than hydrogen bonds, and hydrophobic interaction, to maintain the gel network [28,60].

#### 3.2.3. Microstructure

SEM was examined to elucidate the effect of various HIU pretreatment times on the 3D network structure and morphological changes of TGase-induced 7S gel. The network structure of the untreated group (Figure 5A) was more loose and uneven than that of the treated gels (Figure 5B–E). Gels with HIU pretreatment showed a relatively compact network structure, especially for the gels treated with sonication for 30 min (Figure 5C), which formed a distinct filamentous structure and uniform pores. Notably, with increasing HIU time (>30 min), the gels formed a more discontinuous and irregular, as well as heterogeneous, network structure, in which aggregation even occurred.

The reason for this trend at 30 min was mainly related to the decrease in 7S particle size, as well as the unfolding of the 7S structure under moderate HIU time. In turn, more hydrophobic groups and enzymatic reaction sites were exposed, resulting in enhanced hydrophobic interactions and an increased amount of the ε-(γ-glutamyl)lysine isopeptide bond, which allowed a regular and dense 3D network structure to come into being [61]. This microstructure analysis was in consistent with the intermolecular force analysis results (Section 3.2.2). However, with the increase in HIU pretreatment time, the 7S molecules showed clustering and increased particle size, due to excessive ultrasound cavitation. This was not conducive to cross-linking with TGase into a dense gel network, and in turn resulted in a rough and inhomogeneous network structure. The SEM analyses in this experiment were consistent with the results reported by Hu, et al. [44] and Jiang, et al. [46]. Furthermore, the spatial structure of the gels was closely related to their functional properties [62]. Therefore, the functional properties of gels possessing different protein structures and gel structures are discussed in the next sections.

### 3.3. Functional Properties of TGase-Induced 7S Gels

#### 3.3.1. Gel Strength and Water-Holding Capability (*WHC*) of Gel

Figure 6A shows the effects of HIU pretreatment on gel strength and *WHC*, which both significantly increased in all HIU-treated gel samples, compared to the untreated gels (*p* < 0.05). Meanwhile, Hu, et al. [6] and Zhao, et al. [26] also pointed out similar findings, showing that HIU treatment could lead to an improvement in strength and *WHC* in gels induced by CaSO_4_ and acid, respectively. It was observed an increasing trend within the first 30 min of HIU pretreatment, whereas there was a significant decrease with further pretreatment (>30 min) (*p* > 0.05). After 30 min HIU pretreatment, those two properties (gel strength and *WHC*) reached the highest levels, being approximately 1.54-fold and 1.23-fold higher than those of the untreated samples, respectively.

These enhancements of gel strength and *WHC* were mainly attributed to the formation of the compact network gel structure. Generally, compared to gels with rough and irregular network structures, gels with uniform, fine, and dense pore sizes had a relatively higher gel strength and could bind more water [63], as is consistent with the results of the SEM (Section 3.2.3). In addition, this might also be because of the reduction in protein particle size by sonication, which allowed the protein molecular structure to be unfolded. Furthermore, this unfolding contributed to the exposure of internal groups in 7S, increasing the accessibility of protein–protein and protein-enzyme crosslinks via hydrophobic interactions and ε-(γ-glutamyl)lysine isopeptide bonds [6,60], as demonstrated by solubility analysis (Section 3.2.2). These changes enhanced the three-dimensional network structures of the 7S gel, thereby enhancing the strength and moisture-retaining ability of the gel. Another reason was the formation of a more β-sheet structure after sonication, as demonstrated in Section 3.1.5. The β-sheet was an ordered secondary structure of proteins, which was closely linked to the uniform and fine network structures [64]. By contrast, both properties of the 7S gels performed the lowering trend with the further HIU pretreatment (>30 min). The reason might be attributed to the fact that extreme HIU pretreatment produced excessive acoustic cavitation in the 7S molecule, which caused a slight increment in 7S particle size as well as a slight reduction in hydrophobicity. These factors resulted in protein aggregation and inhibited the exposure of enzymatic catalytic sites, thus possibly leading to the formation of more uneven microstructures.

#### 3.3.2. Thermal Properties Analysis

The impact of different HIU pretreatment times on the thermal properties of 7S gel samples was analyzed using DSC. The thermal property curves of all gels showed a clear peak of heat absorption from 80 to 90 °C (Figure 6B). The position of the heat absorption peak was associated with the vaporization of the bound water, and reflected the denaturation temperature of the gel, which was positively correlated to its thermal steadiness and structural stability, i.e., a higher denaturation temperature in proportion to the more thermal and structural stability of the gel [65]. The thermal denaturation temperatures of the HIU-treated groups were 83.37 °C, 89.39 °C, 89.06 °C, and 85.7 °C, with increasing HIU time, which were all higher than those of the un-sonicated gels (82.72 °C). This indicated that the thermal stability of the gel was improved effectively by ultrasound intervention, with the highest denaturation temperature observed at 30 min of HIU pretreatment.

The difference in the maximum denaturation temperature may be due to the different conformational changes of the 7S protein structure with the increasing HIU pretreatment time. Under shorter or longer pretreatment, it caused the too-low or excessive degree of de-folding of 7S, which in turn caused the gel structure made from them to show loosening, resulting in a weak ability for water binding and heat resistance. By contrast, the smaller-particle-sized protein fractions were formed at 30 min of HIU pretreatment, thus prompting the establishment of a stable and ordered gel network structure caused by a powerful cross-linking of TGase with the protein. Thereby, a maximum optimization was established for the 7S gel structure stability. The ordered structure had higher thermal stability than the network with inadequate protein binding, affected the state of water present in the three-dimensional network, and also inhibited the evaporation of bound water. On the other hand, higher thermal stability was typically associated with a higher content of ordered secondary structure, and the results observed using DSC were consistent with changes in the β-sheet (Section 3.1.5) [64].

#### 3.3.3. Rheological Behavior

The rheological behavior variations of the 7S gel at different HIU pretreatment times were reflected by evaluating the energy storage modulus (G′), loss modulus (G″), and loss tangent (tan δ). As shown in Figure 7, the G′ value and G″ value of all 7S gels showed an increasing trend, with the lowest G′ as well as G″ of the gels at 0 min and the highest of these at 30 min. In addition, the G″ was significantly lower than the G′ within each group. The above results suggested that the 7S gels in this study all formed an intact network structure that exhibited elastic solid properties [66]. Madadlou, et al. [67] also noted a similar result when acid-induced casein gels were treated using ultrasound. Moreover, both G′ and G″ after HIU pretreatment were heightened, compared to the untreated group. This indicated that the viscoelasticity of the gels was enhanced, which correlated with the reduction of undenatured protein content in the gel network [63].

In this study, all dispersions formed gels on the rheometer (tan δ > 0.1). In this experiment, with the extension of HIU time, the gel tan δ values started to decrease greatly and then increase, where the lowest gel tan δ values were obtained at 30 min (Figure 7C). Meanwhile, all pretreated gels had lower tan δ values than the untreated group. This indicated that the proper HIU pretreatment provided the possibility of forming gels with an elastic network structure. In the pretreating process, the highest G′ and G″ and lowest tan δ of the gels were obtained at 30 min, which was closely correlated with the gel network structure reaching its strongest point and forming an elastic gel. In addition, at 60 min of HIU pretreatment, the excessive pretreatment resulted in severe aggregation of 7S. This induced the formation of large agglomerates and negatively affected the gel formation, thus presenting lower G′ and G″ values and an elevated tan δ.

### 3.4. Correlation between Protein Conformation and Gel Properties

To further explore the intrinsic link between protein conformation and gel property, the correlation analysis heat map of the above two related indicators was established. As shown in Figure 8, the stronger negative correlations were obtained between gel strength and particle size (r = −0.93) and α-helix (r = −0.994), while there were stronger positive correlations with Ho (r = 0.928) and β-sheet (r = 0.949). *WHC* was strongly negatively correlated with particle size (r = −0.859), α-helix (r = −0.698), β-turn (r = −0.745), and random coil (r = −0.661), while Ho (r = 0.936) and β-sheet (r = 0.851) were positively correlated with *WHC*. The gel thermal denaturation temperature was strongly negatively correlated with particle size (r = −0.745) and α-helix (r = −0.954), and there was a positive correlation with Ho (r = 0.748) as well as β-sheet (r = 0.847). The changes in protein structure under different HIU pretreatment time are closely related to the functional properties of the gels. At the same time, the reduction in protein particle size and the increase in Ho are associated with the unfolding of the protein structure, which facilitated the improvement of the properties of the later-formed gels. Among the protein structures, particle size (*p* < 0.05), Ho (*p* < 0.05), α-helix (*p* < 0.01), and β-sheet (*p* < 0.05) have an extremely significant impact on the gel quality. Of these, the formation of the α-helix negatively affects the functional properties of the gel, while the formation of the β-sheet brings favorable effects.

## 4. Conclusions

HIU pretreatment has important effects on the physicochemical, structural and functional properties of the TGase-induced 7S gels. The structure and gelling properties of 7S were optimized via HIU pretreatment, especially at 30 min. During this period, the particle size was significantly reduced, the surface hydrophobicity was improved, and a gradual ordering of the secondary structure was formed, along with the evacuation of the tertiary structure. In addition, HIU pretreatment induced the formation of ε-(γ-glutamyl)lysine isopeptide bonds that facilitated the maintenance of the structural integrity of the gel. HIU pretreatment for 30 min produced a more uniformly textured and structurally stable gel network, which resulted in the strongest gel strength, superior water-holding property, and higher thermal denaturation temperature of TGase-induced 7S gels, at this time. HIU pretreatment increased the G′ and G″ of the gel and decreased the tan δ, which prompted the formation of a complete gel network structure and exhibited elastic solid properties. Correlation analysis showed that the changes in protein structure were closely related to the functional properties of the gels. Among these, the functional properties of 7S gels were negatively correlated with the particle size and α-helix (*p* < 0.05), while positively correlated with the Ho and β-sheet (*p* < 0.05). Therefore, the protein conformation can be altered by controlling the time of the HIU pretreatment, which in turn improves the structural and functional properties of the gels. Notably, the above results also provide new insights and a theoretical basis for the development and quality assessment of 7S gels with superior properties.

## Figures and Tables

**Figure 1 foods-12-02037-f001:**
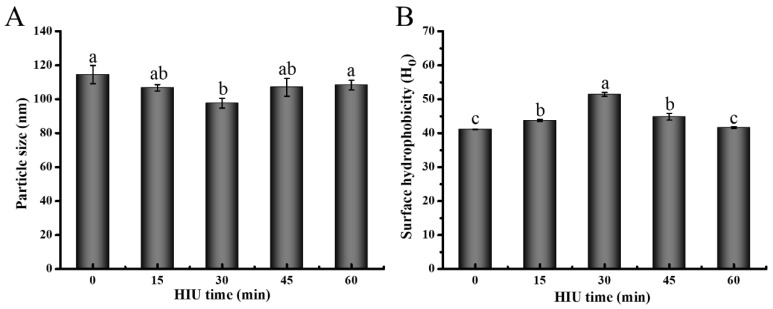
Particle size (**A**) and Ho (**B**) of 7S at different HIU pretreatment times. Different letters above the columns mean significant difference between values (*p* < 0.05).

**Figure 2 foods-12-02037-f002:**
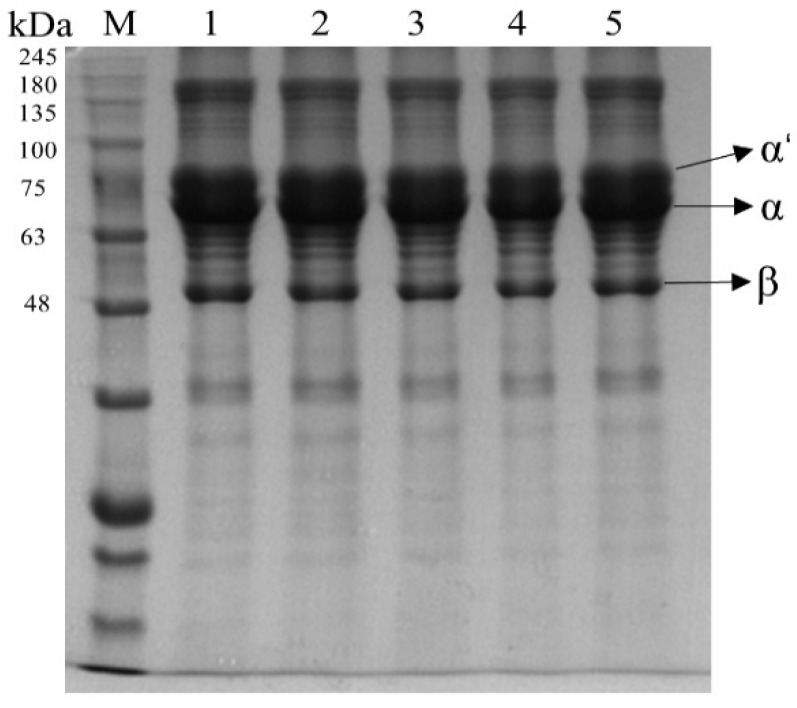
SDS-PAGE electrophoretic profiles of 7S with different high-intensity ultrasound treatment times (M, marker; 1–5, 0; 15; 30; 45; 60 min).

**Figure 3 foods-12-02037-f003:**
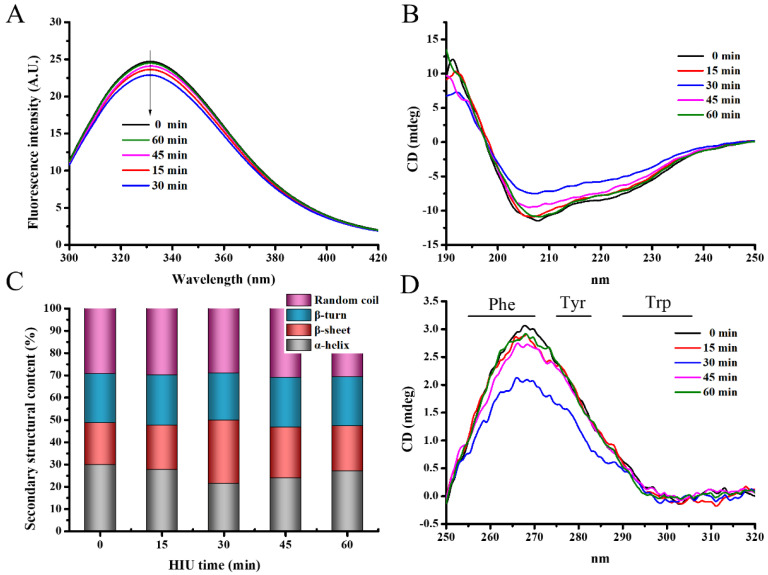
Tryptophan fluorescence spectra (**A**), secondary structure (**B**), secondary structural content (**C**), and tertiary structure (**D**) of 7S at different high-intensity ultrasound pretreatment time.

**Figure 4 foods-12-02037-f004:**
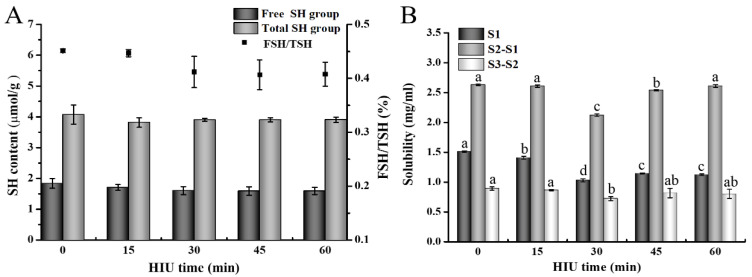
Free sulfhydryl (FSH) and total sulfhydryl (TSH) content (**A**) and solubility of 7S gels (**B**) at different high-intensity ultrasound pretreatment times. Solubility determined in 0.6 M NaCl (S1), 0.6 M NaCl + 8 M urea (S2), and 0.6 M NaCl + 8 M urea + 0.5 M β-mercaptoethanol (S3) (S1, nonspecific associations; S2-S1, hydrogen bonds and hydrophobic interaction; S3-S2, disulfide bonds). Different letters above the column mean significant difference between values (*p* < 0.05).

**Figure 5 foods-12-02037-f005:**
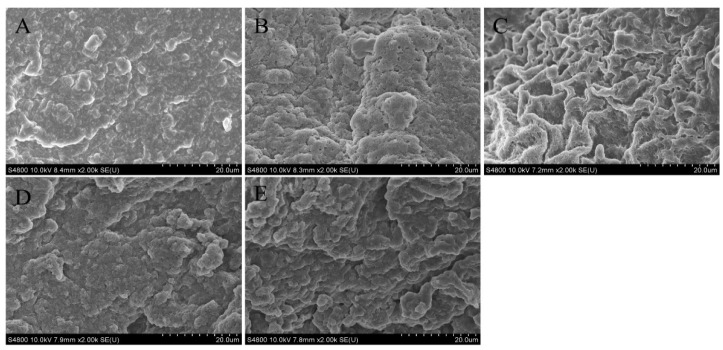
Scanning electron microscopy images of 7S gel at different high-intensity ultrasound pretreatment times ((**A**–**E**) represents the gels prepared from sonication for 0, 15, 30, 45, and 60 min, respectively).

**Figure 6 foods-12-02037-f006:**
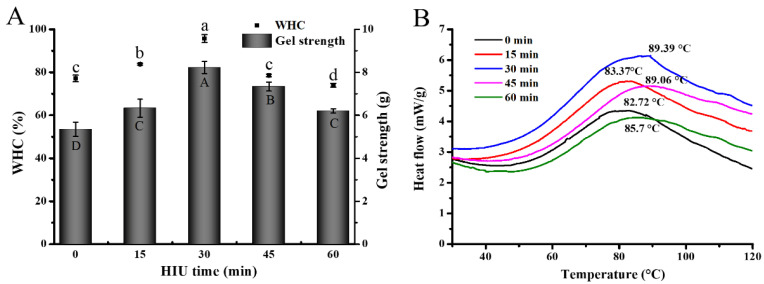
Gel strength and water-holding capability (*WHC*) (**A**) and DSC curves (temperature range: 26–120 °C) (**B**) of gel at different high-intensity ultrasound pretreatment times. Different letters (a–d or A–D) in the graph are significantly different (*p* < 0.05).

**Figure 7 foods-12-02037-f007:**
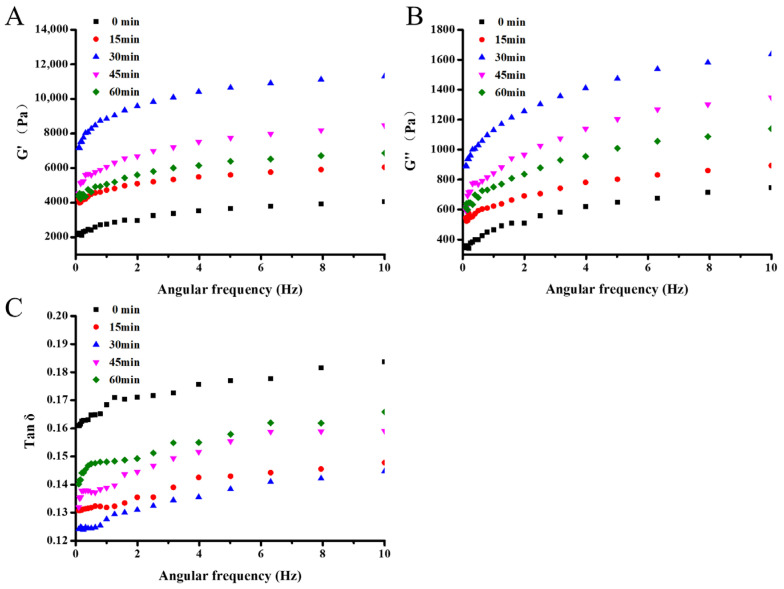
Frequency dependence of the G′ (**A**), G″ (**B**), and tan δ (**C**) of 7S gel at different high intensity ultrasound pretreatment times (G′, storage modulus; G″, loss modulus; tan δ (G″/G′), phase angle).

**Figure 8 foods-12-02037-f008:**
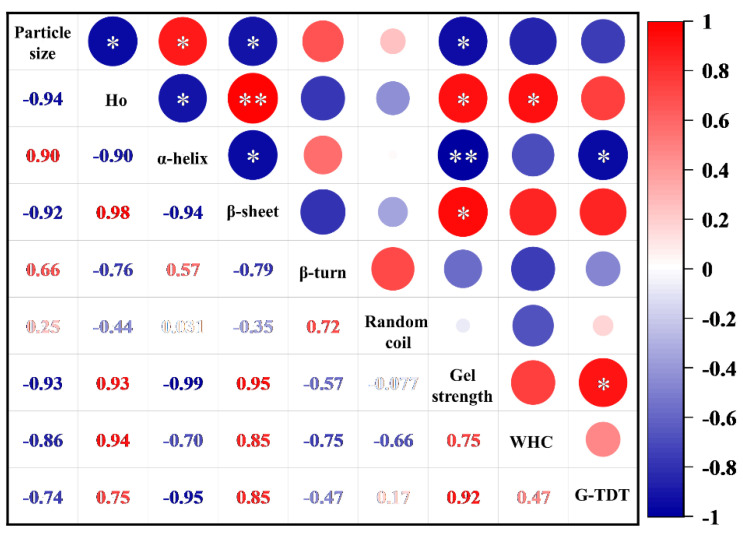
Correlation between protein conformation and gel properties at different high-intensity ultrasound pretreatment times (*WHC*, water-holding capacity; G-TDT, gel thermal denaturation temperature. * indicates significant correlation (*p* < 0.05); ** indicates extremely significant correlation (*p* < 0.01)).

## Data Availability

The data presented in this study are available on request from the corresponding author.

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
