# Peer review of "Effect of High-Intensity Ultrasound Pretreatment on the Properties of the Transglutaminase (TGase)-Induced β-Conglycinin (7S) Gel"

_foods, 2023, doi:10.3390/foods12102037_

Round 1

Reviewer 1 Report

Totally speaking, this article is regarding the improvement of the physicochemical properties and structure of transglutaminase-induced β-conglycinin gels via high-intensity ultrasound pretreatment. The current study's goal was to modify β-conglycinin protein structure by various ultrasound pretreatment times, from 0 to 60 min, and thereby analyze their structural and gel properties induced by transglutaminase. The manuscript was suitable for Foods journal, and proposed Section. However, there are some main points that require clarification.

(1) Affiliation section: Please include the matching e-mail addresses of the co-authors listed above, as well as their initials in parenthesis. Please, see the instructions for authors.

(2) Title section and whole manuscript: I suggest that the authors use the term "high-intensity ultrasound" or "sonication technology", no "ultrasonic", because it has been proven that only high-intensity, low-frequency ultrasound can lead to structural changes in proteins at the quaternary and tertiary structure levels.

(3) Abstract section

There is a lack of a clearly written research goal, the reason for this kind of research, as well as the main objectives of the research. The methodology needs to be rewritten, specifically specifying which analysis was used to determine the gelation properties of proteins and which to determine the structural changes caused by the applied ultrasound treatment. To say something about the treatment used, why it is used, where it is usually used, and what the purpose of its application is. Also, the results must be clearly shown, and what values were obtained. It is necessary to amend and supplement the Abstract section.

(4) Keywords

Please change and include more keywords that are more crucial for the research itself. These are far too broad. So use more precise terms in your research to make it more focused. For example: β-conglicinin; soy proteins; ultrasound treatment; ultrasound time; protein structure; water-holding activity; transglutaminase-induced gel; gel properties; etc.

(5) Introduction section

Please complete the introduction with specific statistical data on the worldwide utilization and production whole soybean protens and isolated soy protein fraction (β-conglicinin), as well as its products. The nutritional composition of soy protein fractions must be listed in detail, including the citation of the necessary literature. Next, isolate the procedures for obtaining soy protein fractions with improved techno-functional and structural chracteristics (i.e., a few illustrations of how the authors enhanced the characteristics of proteins using ultrasonic probes, microwaves, or high pressure).

Please provide a few more research goals at the conclusion of the introductory section. These goals will be divided into phases based on the results of the carried out experimental studies. Hence, the primary goal of this study will be made clear.

(6) Materials and Methods section

A section related to the preparation of 7S protein: Rewrite the precipitate's washing method in writing. Rewrite the drying conditions and the powder storage strategy.

The names of the devices (centrifuge and appropriated rotors; freeze-dryer, etc.) that were used in the whole experimental work and their manufacturers must be mentioned.

How did you make sure the sample stayed cool while being treated with ultrasound using the probe? In order to ascribe structural changes to either ultrasound waves or to the interplay of temperature and ultrasonication, the temperature rises as a result of cavitation, hence it is crucial to maintain the desired temperature. Please, write necessary.

Determination of water holding capacity: Simply using the number 100 in the equation suffices; the percentage need not be included. A mark of the size that is calculated receives the percentage.

(7) Results and Discussion

Perhaps the following chapters could be used to separate the findings discussion:

3.1 Results and analysis from Chapters 3.1, 3.2, 3.3, 3.4, and 3.5 will be used to analyze the impact of ultrasound treatment on protein structural alterations

3.2 Effect of ultrasound treatment on gels properties that will include your results and discussion from Chapters 3.6, 3.7 i 3.9

3.3 Structure properties of gels (chepters 3.8, 3.10 and 3.11)

(8) Weather is conceivable for the authors to rephrase their conclusion section by drawing a general (concluded) influence on each examined propeties (techno-functional, structural, microstructural and gelling properties) of modified protein?

(9) It is advised that the authors recheck the main text during the revision to make this manuscript more readable.

Author Response

Response:

Your and the reviewers’ comments on our manuscript “Effect of high-intensity ultrasonic pretreatment on the properties of the transglutaminase (TGase)-induced β-conglycinin (7S) gel” (Manuscript ID: foods-2363222) are highly appreciated. We have gone through the manuscript again and revised it carefully according to the suggestions. All the changes have been highlighted in red in the revised manuscript. Thus, the revised version is submitted online for you to reconsider for publication in Foods. In the following pages are our point-by-point responses to each of the comments.

Manuscript Number: foods-2363222

Reviewer #1:

Comments:

Totally speaking, this article is regarding the improvement of the physicochemical properties and structure of transglutaminase-induced β-conglycinin gels via high-intensity ultrasound pretreatment. The current study's goal was to modify β-conglycinin protein structure by various ultrasound pretreatment times, from 0 to 60 min, and thereby analyze their structural and gel properties induced by transglutaminase. The manuscript was suitable for Foods journal, and proposed Section. However, there are some main points that require clarification.

(1) Affiliation section: Please include the matching e-mail addresses of the co-authors listed above, as well as their initials in parenthesis. Please, see the instructions for authors.

Response: I would like to express my appreciation for your review of this paper. We have listed the matching e-mail addresses of the co-authors and their initials, as follows: “mx120211221@stu.yzu.edu.cn (L.Z.), mx120221256@stu.yzu.edu.cn (J.Z.), mx120201185@stu.yzu.edu.cn (P.W.), 008122@yzu.edu.cn (J.X.), gycui@yzu.edu.cn (G.C.), 007232@yzu.edu.cn (J.W.)”

(2) Title section and whole manuscript: I suggest that the authors use the term "high-intensity ultrasound" or "sonication technology", no "ultrasonic", because it has been proven that only high-intensity, low-frequency ultrasound can lead to structural changes in proteins at the quaternary and tertiary structure levels.

Response: We have corrected the expression "ultrasonic" in title section and whole manuscript, such as "Effect of high-intensity ultrasound pretreatment on the properties of the transglutaminase (TGase)-induced β-conglycinin (7S) gel" (Line 1); " Modifications of food products by sonication technology include low-intensity ultrasound pretreatment and high-intensity ultrasound pretreatment. The low-intensity ultrasound (5–10 kHz, 1 W/cm−2) …" (Lines 70-72); " High-intensity ultrasound (HIU) is used extensively in food processing " (Line 75).

(3) Abstract section

There is a lack of a clearly written research goal, the reason for this kind of research, as well as the main objectives of the research. The methodology needs to be rewritten, specifically specifying which analysis was used to determine the gelation properties of proteins and which to determine the structural changes caused by the applied ultrasound treatment. To say something about the treatment used, why it is used, where it is usually used, and what the purpose of its application is. Also, the results must be clearly shown, and what values were obtained. It is necessary to amend and supplement the Abstract section.

Response: Thank you for your kind suggestion. We have added the relevant information in the Abstraction section, as follows: “In this study, we investigated the effects of different high-intensity ultrasound (HIU) pretreat-ment time (0-60 min) on the structure of β-conglycinin (7S) and the structural and functional properties of 7S gels induced by transglutaminase (TGase). Analysis of 7S conformation revealed that 30 min HIU pretreatment significantly induced the unfolding of 7S structure with the small-est particle size (97.59 nm), the highest surface hydrophobicity (51.42), and the lowering and raising content of α-helix and β-sheet, respectively. Gel solubility showed that HIU facilitated the formation of ε-(γ-glutamyl)lysine isopeptide bonds that maintain the stability and integrity of the gel network. The SEM revealed that the three-dimensional network structure of the gel at 30 min exhibited filamentous and homogeneous properties. Among them, the gel strength and wa-ter-holding capacity were approximately 1.54 and 1.23 times higher than those of the untreated 7S gels, respectively. 7S gel obtained the highest thermal denaturation temperature (89.39 ℃), G', and G'', and the lowest tan δ. Correlation analysis demonstrated that the gel functional properties were negatively correlated with particle size and α-helix, while positively with Ho and β-sheet. By contrast, gels without sonication or with excessive pretreatment showed a large pore size and inhomogeneous gel network, and poor properties. These results will provide a theoretical basis for the optimization of HIU pretreatment conditions during TGase-induced 7S gel formation to improve gelling properties.” (Lines 12-27)

(4) Keywords

Please change and include more keywords that are more crucial for the research itself. These are far too broad. So use more precise terms in your research to make it more focused. For example: β-conglicinin; soy proteins; ultrasound treatment; ultrasound time; protein structure; water-holding activity; transglutaminase-induced gel; gel properties; etc.

Response: Based on your suggestion, we have supplemented the Keywords, as follows: "Ultrasound pretreatment; Ultrasound time; Soy proteins; β-Conglycinin; Protein structure; Transglutaminase-induced gel; Gelling properties" (Lines 28-29)

(5) Introduction section

Please complete the introduction with specific statistical data on the worldwide utilization and production whole soybean protens and isolated soy protein fraction (β-conglicinin), as well as its products. The nutritional composition of soy protein fractions must be listed in detail, including the citation of the necessary literature. Next, isolate the procedures for obtaining soy protein fractions with improved techno-functional and structural chracteristics (i.e., a few illustrations of how the authors enhanced the characteristics of proteins using ultrasonic probes, microwaves, or high pressure).

Please provide a few more research goals at the conclusion of the introductory section. These goals will be divided into phases based on the results of the carried out experimental studies. Hence, the primary goal of this study will be made clear.

Response: Thank you for your comments. According to your suggestion, we adjusted the Introduction section, such as “Soy protein is used as a crucial food processing aid because it contains high-quality protein required by the human body and is a great source of essential fatty acid and isoflavone [1]. In addition, it has a superior ability to improve the texture of food. According to statistics, the global production of soy protein is estimated to be 392 million tons in 2022. Gel products are the largest component, with a market share of 41% [2].” (Lines 32-36); “The study has shown that 7S has excellent physiological functions such as enhancing LDL metabolism and lowering triglycerides in human hepatocytes [4].” (Lines 38-40); “Feng, et al. [9] stated that the conventional thermally induced 7S gels had a loose and brittle structure and exhibited poor functional properties. Kohyama et al. [10] also mentioned that 7S gels possessed lower levels of fracture stress values and higher shrinkage, which is not favorable for the practical application of the gels.” (Lines 43-46); “Sun, et al. [14] demonstrated that the strength of pea protein and soy protein gels induced with TGase was 8-fold and 2-fold higher than that of the control, respectively. Similarly, Chen, et al. [15] and Qin, et al. [16] noted that the addition of TGase significantly promoted the formation of fine gel structures of rapeseed isolate and wheat glutenin, which in turn improved the gel properties.” (Lines 58-62); “For example, whey [22] and quinoa [23] protein showed significant unfolding of protein structures and the increasing solubility of the protein dispersions after sonication. Li, et al. [24] found that the α-helix and β-sheet content in the protein secondary structure was decreased and increased, respectively, which was related to the de-folding of the protein conformation, when they prepared peanut isolate-polysaccharide couples by ultrasonication combined with glycosylation reaction.” (Lines 79-84); “Therefore, this study investigated the effect of different HIU pretreatment time on the 7S structure and the gel properties induced by TGase. The changes of 7S conformations (primary, secondary, tertiary, and quaternary structures) with different HIU pretreatment time were compared. The differences in structural properties as well as functional properties of gels possessing these structural variations of protein are explored. The correlation between the protein conformation and the functional properties of the gel was analyzed.” (Lines 97-102)

(6) Materials and Methods section

A section related to the preparation of 7S protein: Rewrite the precipitate's washing method in writing. Rewrite the drying conditions and the powder storage strategy. The names of the devices (centrifuge and appropriated rotors; freeze-dryer, etc.) that were used in the whole experimental work and their manufacturers must be mentioned.

Response: The method related to the preparation of 7S protein was rewritten and the names of the devices and their manufacturers have been added. The detailed explanation had been supplemented, as follows: “…was centrifuged (9,000 g, 30 min, 4 °C) using the Allegra X-30R Centrifuge (F0850 rotor, Beckman Coulter, Inc., USA).” (Lines 118-119); “The precipitate was washed 3 times with distilled water (DIW) and the pH was adjusted to 7.5 with 2M NaOH. After that, the precipitate was freeze-dried using a Telstar (Lyoquest-55) laboratory freeze dryer (Nanjing Xinfeida Optoelectronics Science and Technology Co., Ltd, Nanjing, China) for 48 h at −55 °C to obtain the 7S” (Lines 127-131)

How did you make sure the sample stayed cool while being treated with ultrasound using the probe? In order to ascribe structural changes to either ultrasound waves or to the interplay of temperature and ultrasonication, the temperature rises as a result of cavitation, hence it is crucial to maintain the desired temperature. Please, write necessary.

Response: Thanks for your reminder, the relevant information has been added to the manuscript, as follows: “Throughout the HIU pretreatment phase, the flat-bottom conical flask containing the solution was placed in an ice water bath.” (Lines 137-138)

Determination of water holding capacity: Simply using the number 100 in the equation suffices; the percentage need not be included. A mark of the size that is calculated receives the percentage.

Response: We have corrected the problem, as follows: “…The WHC was calculated using the following formula [36]:

WHC (%)=(m-(m1-m2 ) )/m×100

where m (g) is the total mass of the gel before centrifugation, m1 (g) is the total mass of the centrifuge tube and gel before centrifugation, and m2 (g) is the total mass of the centrifuge tube and gel after centrifugation.” (Lines 226-230)

(7) Results and Discussion

Perhaps the following chapters could be used to separate the findings discussion:

3.1 Results and analysis from Chapters 3.1, 3.2, 3.3, 3.4, and 3.5 will be used to analyze the impact of ultrasound treatment on protein structural alterations

3.2 Effect of ultrasound treatment on gels properties that will include your results and discussion from Chapters 3.6, 3.7 i 3.9

3.3 Structure properties of gels (chepters 3.8, 3.10 and 3.11)

Response: We appreciate it very much for this good suggestion, and we have done it according to your ideas, as follows:

“3.1. Conformational characteristics of 7S

3.1.1. Particle size

3.1.2. Surface hydrophobicity (Ho)

3.1.3. SDS-PAGE

3.1.4. Intrinsic fluorescence spectra

3.1.5. Circular dichroism (CD) spectra

3.2. Structure properties of TGase-induced 7S gels

3.2.1. Free sulfhydryl (FSH) and total sulfhydryl (TSH) contents of gel

3.2.2. Solubility of gel

3.2.3. Scanning electron microscopy (SEM)

3.3. Functional properties of TGase-induced 7S gels

3.3.1. Gel strength and water holding capability (WHC) of gel

3.3.2. Thermal properties analysis

3.3.3. Rheological behavior

3.4. Correlation between protein conformation and gel properties”

(8) Weather is conceivable for the authors to rephrase their conclusion section by drawing a general (concluded) influence on each examined propeties (techno-functional, structural, microstructural and gelling properties) of modified protein?

Response: Thank you for the suggestion. we have rewritten the Conclusion section as shown below.

“HIU pretreatment has important effects on the physicochemical, structural and functional properties of the TGase-induced 7S gels. The structure and gelling properties of 7S were optimized via HIU pretreatment, especially at 30 min. During this period, the particle size was significantly reduced, the surface hydrophobicity was improved, and a gradual ordering of the secondary structure was formed, along with the evacuation of the tertiary structure. In addition, HIU pretreatment induced the formation of ε-(γ-glutamyl)lysine isopeptide bonds that facilitated the maintenance of the structural integrity of the gel. HIU pretreatment for 30 min produced a more uniformly textured and structurally stable gel network, which resulted in the strongest gel strength, superior water-holding property, and higher thermal denaturation temperature of TGase-induced 7S gels at this time. HIU pretreatment increased the G’ and G’’ of the gel and decreased tan δ, which prompted the formation of a complete gel network structure and exhibited elastic solid properties. Correlation analysis showed that the changes in protein structure were closely related to the functional properties of the gels. Among them, the functional proper-ties of 7S gels were negatively correlated with the particle size and α-helix (p<0.05), while positively correlated with the Ho and β-sheet (p<0.05). Therefore, the protein conformation can be altered by controlling the time of HIU pretreatment, which in turn improves the structural and functional properties of the gels. Notably, the above results also provide new insights and a theoretical basis for the development and quality assessment of 7S gels with superior properties.” (Lines 572-590)

(9) It is advised that the authors recheck the main text during the revision to make this manuscript more readable.

Response: Thank you for your valuable and thoughtful comments. We have carefully checked and improved the English writing in the revised manuscript.

Reviewer 2 Report

The manuscript represents a new contribution, consistently and correctly performed that will be of interest to the researchers. The study is interesting and has novelty. The manuscript has been written and presented very well. The introduction sets the manuscript in an international context and shows how it builds on previous work on the subject. The research has been very well designed and executed. The methods of analysis are correct and well presented.  The data collected were appropriately analyzed and the results are clearly and properly treated and well presented in figures. The discussion is appropriate in terms of scope and content and very well supports the findings and justifies them

Author Response

Response:

Your and the reviewers’ comments on our manuscript “Effect of high-intensity ultrasonic pretreatment on the properties of the transglutaminase (TGase)-induced β-conglycinin (7S) gel” (Manuscript ID: foods-2363222) are highly appreciated. We have gone through the manuscript again and revised it carefully according to the suggestions. All the changes have been highlighted in red in the revised manuscript. Thus, the revised version is submitted online for you to reconsider for publication in Foods. In the following pages are our point-by-point responses to each of the comments.

Manuscript Number: foods-2363222

Reviewer #2:

Comments:

The manuscript represents a new contribution, consistently and correctly performed that will be of interest to the researchers. The study is interesting and has novelty. The manuscript has been written and presented very well. The introduction sets the manuscript in an international context and shows how it builds on previous work on the subject. The research has been very well designed and executed. The methods of analysis are correct and well presented.  The data collected were appropriately analyzed and the results are clearly and properly treated and well presented in figures. The discussion is appropriate in terms of scope and content and very well supports the findings and justifies them.

Response: We would like to express our appreciation for your review of this paper. We would like to thank you again for your recognition of our work. We will continue to do our best. Sending you our most sincere wishes.

Reviewer 3 Report

The paper Effect of high-intensity ultrasonic pretreatment on the properties of the transglutaminase (TGase)-induced β-conglycinin (7S) gel is interesting and worth of investigation. However, I have some remarks:

Authors described and discussed results separately. It is crucial to find the relation between selected analysed parameters. I suggest to perform the more detailed statistical analysis e.g. the correlation analyse or PCA.  Analysis of these data will make it possible to link the obtained data and to discuss the results in a broader way.

The article requires linguistic and editorial correction e.g.

-page 1 line 4 of Introduction “interacti

-page 2 “peotein” should be protein

In materials and methods:

2.16 Rheological properties: Why the samples where heated from 25 to 90C and cooled.   Did authors measure the gelation temperature of samples?

There are some typos in the article that need to be corrected  

Author Response

Response:

Your and the reviewers’ comments on our manuscript “Effect of high-intensity ultrasonic pretreatment on the properties of the transglutaminase (TGase)-induced β-conglycinin (7S) gel” (Manuscript ID: foods-2363222) are highly appreciated. We have gone through the manuscript again and revised it carefully according to the suggestions. All the changes have been highlighted in red in the revised manuscript. Thus, the revised version is submitted online for you to reconsider for publication in Foods. In the following pages are our point-by-point responses to each of the comments.

Manuscript Number: foods-2363222

Reviewer #3:

Comments:

The paper Effect of high-intensity ultrasonic pretreatment on the properties of the transglutaminase (TGase)-induced β-conglycinin (7S) gel is interesting and worth of investigation. However, I have some remarks:

Authors described and discussed results separately. It is crucial to find the relation between selected analysed parameters. I suggest to perform the more detailed statistical analysis e.g. the correlation analyse or PCA.  Analysis of these data will make it possible to link the obtained data and to discuss the results in a broader way.

Response: Thanks for your suggestion, the relevant information has been added to the manuscript, as follows:

“3.4 Correlation between protein conformation and gel properties

To further explore the intrinsic link between protein conformation and gel property, the correlation analysis heat map of the above two related indicators was established. As shown in Fig. 8, the stronger negative correlations were obtained between gel strength and particle size (r = -0.93) and α-helix (r = -0.994), while there were stronger positive correla-tions with Ho (r = 0.928) and β-sheet (r = 0.949). WHC was strongly negatively correlated with particle size (r = -0.859), α-helix (r = -0.698), β-turn (r = -0.745), and random coil (r = -0.661), while Ho (r = 0.936) and β-sheet (r = 0.851) were positively correlated with WHC. The gel thermal denaturation temperature was strongly negatively correlated with particle size (r = -0.745) and α-helix (r = -0.954), and there was a positive correlation with Ho (r = 0.748) as well as β-sheet (r = 0.847). The changes in protein structure under different HIU pretreatment time are closely related to the functional properties of the gels. At the same time, the reduction of protein particle size and the increase of Ho are associated with the unfolding of protein structure, which facilitated the improvement of the properties of the later-formed gels. Among the protein structures, particle size (p<0.05), Ho (p<0.05), α-helix (p<0.01), and β-sheet (p<0.05) have an extremely significant impact on the gel quality. Of these, the formation of the α-helix negatively affects the functional properties of the gel, while the formation of the β-sheet brings favorable effects.

Figure 8. Correlation between protein conformation and gel properties at different high-intensity ultrasound pretreatment time (WHC, water-holding capacity; G-TDT, gel thermal denaturation temperature. * indicates significant correlation (p < 0.05); ** indicates extremely significant correlation (p < 0.01))” (Lines 547-570)

The article requires linguistic and editorial correction e.g.

-page 1 line 4 of Introduction “interacti”

-page 2 “peotein” should be protein

Response: I would like to express my appreciation for your review of this paper. We have corrected the relevant problems in the manuscript, such as “The β-conglycinin (7S), an important component of soy protein, is a trimer composed of three subunits (α′, α, β), linked by non-covalent interactions” (Lines 36-38); “The textural characteristics of soy protein gels induced by CaSO4....” (Lines 84-85). We have also reviewed and revised the entire text, which has been highlighted in red in the manuscript.

In materials and methods:

2.16 Rheological properties: Why the samples where heated from 25 to 90C and cooled.   Did authors measure the gelation temperature of samples?

Response: Dear Reviewer, here is my response to this question. During the experiment, we found that the heating and cooling procedure set up above worked well to cause the gel forming solution to form gels on the bench. Another effect of this procedure was to inactivate transglutaminase. Subsequently cooling to room temperature was used to measure the rheological properties of the gels. Prior to the start of all our experiments, a number of pre-experiments were performed to determine the gel-forming temperature range (20-90°C) of the TGase-induced 7S protein by referring to the literature of Feng, et al. [1], Zhang, et al. [2], Li, et al. [3].

References:

[1] Feng G., Liu W.H., Chen Z.G.. Analysis of the structure and thermal gel properties of soybean 7S and 11S proteins [J]. food science, 2020,41 (02): 58-64

[2] Zhang Q., Wu H.B., Yan W.W., et al. Research progress on the physicochemical properties and modification of soybean 7S and 11S globulins [J]. Food and Fermentation Industry, 2022,48 (09): 324-335

[3] Li Y., Zhong F., Ma J.G. Gel properties of soybean protein components 7S and 11S [J]. Journal of Food and Biotechnology, 2005 (06): 19-23

Comments on the Quality of English Language

There are some typos in the article that need to be corrected

Response: Thanks for your reminder, we have carefully checked the manuscript and corrected the typos

Round 2

Reviewer 3 Report

The authors responded to the comments and significantly improved the article